# A Proposal for a Consolidated Structural Model of the CagY Protein of *Helicobacter pylori*

**DOI:** 10.3390/ijms242316781

**Published:** 2023-11-26

**Authors:** Mario Angel López-Luis, Eva Elda Soriano-Pérez, José Carlos Parada-Fabián, Javier Torres, Rogelio Maldonado-Rodríguez, Alfonso Méndez-Tenorio

**Affiliations:** 1Laboratorio de Biotecnología y Bioinformática Genómica, Departamento de Bioquímica, Escuela Nacional de Ciencias Biológicas, Instituto Politécnico Nacional, Campus Lázaro Cárdenas, Mexico City 11340, Mexico; mlopezl1405@alumno.ipn.mx (M.A.L.-L.); esorianop1600@alumno.ipn.mx (E.E.S.-P.); jparadaf@ipn.mx (J.C.P.-F.); romaldonado@ipn.mx (R.M.-R.); 2Unidad de Investigación en Enfermedades Infecciosas, UMAE Pediatría, Instituto Mexicano del Seguro Social, Mexico City 06720, Mexico; uimeip@gmail.com

**Keywords:** bioinformatics, structural biology, CagY protein, T4SS, deep learning

## Abstract

CagY is the largest and most complex protein from *Helicobacter pylori’s* (Hp) type IV secretion system (T4SS), playing a critical role in the modulation of gastric inflammation and risk for gastric cancer. CagY spans from the inner to the outer membrane, forming a channel through which Hp molecules are injected into human gastric cells. Yet, a tridimensional structure has been reported for only short segments of the protein. This intricate protein was modeled using different approaches, including homology modeling, ab initio, and deep learning techniques. The challengingly long middle repeat region (MRR) was modeled using deep learning and optimized using equilibrium molecular dynamics. The previously modeled segments were assembled into a 1595 aa chain and a 14-chain CagY multimer structure was assembled by structural alignment. The final structure correlated with published structures and allowed to show how the multimer may form the T4SS channel through which CagA and other molecules are translocated to gastric cells. The model confirmed that MRR, the most polymorphic and complex region of CagY, presents numerous cysteine residues forming disulfide bonds that stabilize the protein and suggest this domain may function as a contractile region playing an essential role in the modulating activity of CagY on tissue inflammation.

## 1. Introduction

*Helicobacter pylori* (Hp) is a bacterium that colonizes the stomach of over 50% of the human population and has evolved to specifically grow in the harsh environment of the human gastric mucosa [1]. Gastric colonization is possible due to its spiral shape, multiple unipolar flagella, and to the production of urease that counteracts the extreme acidity of the stomach [2,3].

Hp is usually acquired during early childhood, probably by intimate oral–oral contact with the mother [4] to co-exist with human gastric cells for life in most cases (over 80%), representing a clear example of microbiota vertically transmitted from mother to child. However, in a few cases, Hp might cause peptic ulcers and may become a risk factor for gastric cancer (GC) [5]. Hp strains with increased capacity to cause GC encode virulence genes like those of the Type IV Secretion System (T4SS), adhesins, and a cytotoxin [6,7]. The Hp T4SS translocates the CagA protein, DNA, and heptose into the cytoplasm of gastric cells. CagA has multiple effects in the host cells, activating several pathways, and it was the first bacterial protein to be recognized as an oncoprotein [6,7,8,9,10]. The Hp T4SS is a complex secretion system with partial homology to proteins from other bacterial T4SSs [11,12]. The largest protein in the Hp T4SS is CagY, a protein of about 2000 amino acids (aa) with a carboxylic terminal region homologous to VirB10, but the rest of the protein has no homology with other known proteins [8,9].

The CagY protein is widely variable in length and sequence, and its structure is highly complex, presenting two repeated regions (RR), one known as the 5′ region (FRR) (at the amino terminus), which is an intrinsically disordered region (IDR), and the other in the middle of the protein, the unusually long middle repeat region (MRR) with almost 1000 aa (Figure 1a). The MRR repeats are classified as A and B modules, which are composed of three distinct motifs: delta, mu, and alpha for module A, and epsilon, lambda, and beta for module B [13,14] (Figure 1b). It has been suggested that the variation in the number and location of the repeats in MRR may be involved in the regulation of the translocation of CagA by the T4SS and in the modulating activity of the host immune response. CagY may regulate gastric tissue inflammation by interacting with the Toll-like receptor 5 (TLR5) [15,16,17,18,19,20]. The MRR is susceptible to rearrangements (by modification, deletion, or insertion of modules) that may affect the structure of CagY and hence its function, including modulation of the inflammatory response [21] and the phosphorylation of CagA [22].

However, the complex and large size of CagY makes its study challenging, particularly in a high number of strains. To date, a total of 3446 *cagY*/CagY coding genes/proteins exists in the NCBI database, but most of them are truncated or incomplete (NCBI, 2023), mainly due to problems in short-read sequencing techniques and in assembling the repetitive regions. The recent long-read sequencing technologies (SMRT/PacBio or Oxford-Nanopore) have allowed researchers to get more accurate and reliable full CagY sequences to better study their complexity and possible function.

The complexity of CagY has hindered the elucidation of its tridimensional structure, with only parts of the VirB10 homologous region recently reported [23,24], which represents only a fraction of the protein (approximately 20% of the sequence). Elucidating the structure of CagY and of the whole T4SS is a challenge, but this is essential to better understand the interaction and mechanisms of the function of T4SS proteins. The most recently reported structure, elucidated by cryo-EM, shows the core of the T4SS composed of an outer membrane complex (OMC), a periplasmic ring (PR), and a stalk, where fragments of the proteins CagY (Cag7), CagX (Cag8), CagT (Cag12), CagM (Cag16), and CagD (Cag3) have been identified. The pilus or core of the T4SS was found to be composed mainly of CagY and CagL [25]. These works have helped to partially decipher the assembly of the T4SS; still, most of the tridimensional structure of CagY is unknown. CagY seems to play a major role in the structure and function of the T4SS, probably spanning from the inner Hp membrane to the periplasmic space and to the outer membrane, until facing the host cell and most probably contacting the cell receptors [11,12]. It is then of the utmost importance to clarify the structure of the whole protein and the function of this major protein in the T4SS.

As a complement to the experimental techniques, different bioinformatic methods can be used to model and predict the structure of proteins [26]. For years, the most accurate method for protein structure prediction was homology modeling, where a tridimensional known structure is used as a template for modeling the structure of a close sequence homologue [27]. Until recently, these computer methods had limitations to predict the structure of proteins with a low identity to homologues with a known tridimensional structure. However, with the advent of deep learning approaches such as AlphaFold2, bioinformatic methods for structure prediction reached unprecedented accuracy [28,29]. Threading methods, also known as fold recognition methods, are currently used to search potential low-similarity templates for remote homologues [30]. In ab initio modeling, methods are used to build the structure of fragments from the aa sequence with the help of short tridimensional templates from known structures, or molecular dynamics modeling and posterior simulation and optimization for their assembly [31]. On the other hand, deep learning-based modeling uses neural networks to calculate the tridimensional structure from all available information in structural databases [32]. These methods can be combined to achieve the highest quality and efficiency in protein modeling [33]. Nevertheless, even these methods still have limitations for complex targets such as CagY. In this work, we combine different approaches to elucidate a hypothetical structure for the CagY protein, which may allow us to understand how it interacts with the remaining proteins of the T4SS.

We propose a tridimensional model for most of the CagY protein, based on all available data on the protein and theoretical data presented in multiple works on CagY. We used the available tridimensional models for fragments of CagY in the PDB database for homology modeling, and ab initio methods to model segments of CagY with missing tridimensional structures. Moreover, deep learning methods (DL) were used to predict the structure of the MRR of CagY.

## 2. Results

### 2.1. Building the CagY Model

The CagY protein was divided into different motifs/segments for modeling purposes, as described in Figure 1a and Table 1. The protein was divided into three functional domains: the 5’ repeat region (FRR), the middle repeat region (MRR), and the homologous VirB10 region. The motifs corresponding to the two predicted transmembranal regions and the antenna projections (AP) [34] were highlighted. Figure 1b summarizes the predictions for the secondary structure of the protein using the PsiPred server. Figure 1b shows the logos for modules A and B from the MRR, calculated from the consensus sequences of the repeats present in the CagY sequence.

A model based on the symmetric 14-chain CagY model was constructed and included most of the protein. For the purposes of this work, the suggested asymmetric complex was not considered for the construction of the model.

### 2.2. Modeling of the Homologous VirB10 Region

The first step for building the tridimensional structure was homology modeling of the CagY protein using the sequence from the Hp 26,695 WT strain (sequence accession WP_103414807) and the described 6ODI and 6X6J cryo-EM structures of CagY from the WT strain of Hp as templates [11]. The model was based on the symmetric 6ODI model containing only 14 subunits of CagY and includes the residues from 1677 to 1907 (residues from 1817 to 1849 are missing in this structure), whereas the 6X6J template includes the residues from 1469 to 1603 (Figure 1a). These templates were helpful in the modeling of the regions corresponding to the 6ODI and 6X6J fragments (Figure 2a).

The residues 1817 to 1849, missing in the 6ODI structure, are part of the AP region, a channel-like domain formed by multiple helix–loops [34]. The AP region of CagY is located as a “crown” structure at the upper portion, above the OM [11,12]. Its structure was modeled for the 26,695 WT by homology modeling with SWISS-MODEL, using the 14-chain 6ODI structure as the multimeric template.

The region joining the structures 6ODI and 6X6J, corresponding to residues 1604 to 1677, was also missing in the models. Secondary structure predictions of this connecting segment were compatible with an alpha helix motif (See Appendix A). Independent ab initio modeling of this region with I-TASSER and Robetta yielded similar alpha helix models with the proper dimensions to fit the space between 6ODI and 6X6J, with slight variations in the overall conformation of the strand (See Appendix A). We selected the model provided by the I-TASSER server.

Thus, a monomer model of residues 1469 to 1833 was calculated using a single chain of both 6ODI and 6X6J, and the I-TASSER model as templates (Appendix A); the assembled model of the three regions is shown in Figure 2b. The resulting model was used to align all the 14 chains of the 6ODI structure, including the AP structure predicted by SWISS-MODEL, and the symmetric multimeric model obtained is shown in Figure 2c. Interestingly, the region corresponding to the 6X6J structure fits a space similar to that filled by the asymmetric 6X6J structure, but with a slightly higher interspace between the subunits, even though this asymmetric structure has 17 alternated units of CagY and CagX.

### 2.3. Modeling of the Middle Repeat Region of CagY

The most challenging region of CagY, which includes part of the transmembrane motif and the MRR (from residues 366 to 1469), was initially modeled with Robetta, I-TASSER, and AlphaFold2/ColabFold. However, the structural conformation of the single chain differed considerably between programs, except for a strong coincidence in the secondary structure (Appendix A). This was similar to the secondary structure predicted by PsiPred for the MRR (Appendix A). The structure predicted with AlphaFold2/ColabFold had considerable differences with the anterior models but received reasonable support, according to its predicted lDDT (local Distance Difference Test) value (Appendix A).

To improve the modeling of the MRR, we calculated a dimer model of this region with AlphaFold2/ColabFold-MMseqs2, which may result in a more efficient prediction of the structure of multimeric proteins [32]. Memory limitations in our equipment prevented the calculation of a high order multimer. Confidence values (predicted lDDT values) for the dimer structure were lower than for the monomer, and it was only possible to calculate two dimer models. However, both models showed remarkable similarities with the monomer, with a more compact packing (Appendix A), and an lDDT value above 60, which implies reasonable confidence for the dimer modeling (Appendix A). In addition, both chains in the dimer had similar conformations and were oriented in parallel from the C to the N terminus (Figure 3a).

By aligning the A and B chains obtained for the best dimer model, we built a trimer ABB′ (Figure 3b), which was subject to an equilibrium molecular dynamics (EMD) simulation of 10 ns in explicit solvent (Figure 3c). The global RMSD values for the trimer showed that its conformational changes did not stabilize during this period. However, a closer examination of the RMSD values for the individual chains showed that chains at the extremes of the model (A and B′) had higher structural variation. In contrast, the central chain structure (B) presented lower variation and showed a tendency to an RMSD equilibrium state. These results suggest that the interactions of the central chain with the neighboring chains are essential for maintaining the conformation in the multimer. In order to demonstrate this, a second EMD was run with a second trimer constructed by replacing all chains with the optimized ones. The behavior of this trimer was similar to that of the previous trimer (Appendix A). An analysis of the changes in the RMSD values per residue for both EMD simulations suggest that during the first EMD simulation there were strong conformational fluctuations for the internal residues, even for the central chain in general (Appendix A). However, the second EMD simulation showed considerably fewer conformational changes (Appendix A), which we consider to be an indicator of a more stable conformation for all chains of the trimer in general. Therefore, we considered the trimer’s central chain (B) as the representative conformation of CagY (Appendix A).

The representative structure of the trimer, optimized by EMD, was used to calculate the stereochemical properties with the PDBsum-EBI server. The analysis suggested that all 58 Cys residues in this region participate in disulfide bond formation (Figure 4a). The PDBSum information for the trimer structure indicated only disulfide intrachain bonds, but there was no support for interchain bonds. A detailed examination of the reported structure analysis showed that the 29 disulfide bonds are homogeneously distributed in the repeated modules, with two Cys residues per module (Figure 4a). Each disulfide bond was consistently placed between a short alpha helix (10 aa) and a consecutive long alpha helix (17 aa), with a spacing of 3 aa. It was interesting to note that the Cys residues were found only in this region of the protein (note their absence in Figure 4b). The Ramachandran plot showed that over 95% of the residues had psi and phi angle values in the most favored regions (Figure 4c). We will delve into the importance of confirming the presence of disulfide bonds in this part of the protein later in the discussion. 

Additional validation tests were performed for the assembled CagY model, with the servers ProSA-web [35], ERRAT [36], QMEANDisCo [37], QMEAN4 [38], and residue contacts with ChimeraX [39]. In all tests the model displayed acceptable quality values (See Appendix A).

### 2.4. Final Assembly of CagY

In the structure of the dimer and trimers, the MRR fragments of CagY were parallelly oriented with the N and C extremes placed in opposite positions (Figure 3a,b). Therefore, joining the optimized central structure of the trimer with the previous model for the VirB10 region was simple, as shown in Figure 5a. Similarly, this structure was aligned with each of the 14 chains of the 6ODI structure, yielding the final model shown in Figure 5b.

We previously mentioned that CagY has several disulfide bonds, restricted to the MRR region, and distributed in the CagY model (Figure 5c). A close view of these Cys residues in the MRR region, as well as a bottom view of the entire CagY complex, are shown in Figure 5c. It is apparent that the series of disulfide bonds were homogeneously distributed, describing five “rings” concentric to the central channel of the multimeric complex. This bell-shaped structure of the ordered component of CagY fits well with previously reported electron microscopy maps (Figure 6) and the cryo-EM structures of the OMC (Figure 7) as described below.

The electron density map for the T4SS OMC of Hp, obtained by single-particle electron microscopy with a resolution of 3.8 Å (EMDB, access EMD-20020), is shown in Figure 6a. The map has a good fit with the structures PDB 6ODI, 6OEG, 6OEE, 6OEF, and 6OEH of the PDB as described by Chung et al., 2019 [11]. The section with the largest volume and high resolution, the upper crown, corresponds to the structures previously reported for the OMC. Furthermore, two diffuse clouds were visible beneath this structure, as also shown in Figure 6a. The best-outlined of these clouds corresponded to the part of the complex that is located below the internal membrane (IM), which includes the potentially disordered region of CagY, the T4SS proteins of the internal membrane, and the energy complexes. Above this region, another less-defined cloud was identified in the area where the missing part of CagY in the OMC structures, which includes the MRR, would be expected to be found. Although the resolution of this technique is not sufficient to calculate the structure of this region, our predicted model fits correctly with it (see Figure 6a, bottom image). It is important to highlight that the cryo-EM map also displays a tunnel/pore in the middle of the image that fits adequately with our model (Figure 6b).

On the other hand, the assembled multimer of CagY was aligned with the cryo-EM structures 6X6S and 6ODI (see Figure 7a), which corresponds to the OMC, PR, and stalk of the Hp 26,695 reference strain [12]. The structures have a good fit, and it is interesting to see that some portions of the MRR in the modeled CagY were at a close distance (about 5–6 Å) from each of the 14 regions of the OMC that authors have reported as unknown fragments, i.e., fragments observed in the experimental maps of 6X6S but not identified at the sequence level (Figure 7b). It is possible that some of these fragments may correspond to portions of CagY in the experimental maps. However, they may also belong to a different protein that mediates the contacts in these regions [12]. 

### 2.5. Changes in the AP Region of CagY Alters the Conformation of the OMC

We also built variants of CagY corresponding to the AP described in a recently published work [34] to investigate if the present model can provide insights regarding the translocation properties of the CagA protein. The AP region corresponds to a loop located between the residues 1820 and 1851, which is not visible in the 6ODI model, where it is reported as a missing structure. This region is flanked by two α-helix motifs that include the residues 1763 to 1863 and was modeled by homology with SWISS-MODEL (Figure 8a) and deep learning with AlphaFold2/ColabFold (Figure 8b). The structure models for the WT sequence look similar with both methods, with only small topologic differences. Then, we modeled different AP sequence variants with both methods. The GS20 model replaces most of the residues in the AP region with Gly and Ser residues. The structure from homology modeling (Figure 8c) displays a crown-like shape. By contrast, the deep learning structure does not show the crown structure (Figure 8d), but the pore diameters are similar in both GS20 models. In the next models the AP from the WT sequence of CagY is replaced with that from *Xanthomonas citri* (Xc), which is a shorter sequence. In this case, the homology model has a large pore (Figure 8e), whereas in the deep learning model the pore diameter is very small (Figure 8f). Finally, the AP was completely deleted in the CagY ΔAP model. SWISS-MODEL produces a structure with several conformational changes with respect to the 6ODI template, which has a relatively large diameter (Figure 8g). It was not possible to model this region with AlphaFold2/ColabFold as the remaining small portions of the structure were arranged randomly. In summary, the deep learning predicts models with pores of smaller diameters when the AP length decreases, while SWISS-MODEL predicts larger pore diameters when the AP length decreases.

## 3. Discussion

The CagY protein is an essential structural component of the T4SS of Hp, with a key participation in its regulation [13,14,20,21,40,41]. The length of the protein from the Hp 26,695 reference strain is 1927 amino acids and has unusually long repetitive elements (close to 900 amino acids) that might be related to their immunogenicity and pathogenicity properties [15,20,42]. It has been reported that CagY completely spans the cell envelope of Hp from the internal membrane (IM) to the outer membrane (OM). This arrangement agrees with bioinformatic predictions of the secondary structure of CagY, which show two transmembrane regions (Figure 1a and Appendix A). Considering its length and extension through the membrane, it is suggested that CagY plays an essential role in the translocation of the CagA oncoprotein [11,12,43].

CagY is an unusual molecule that seems to have evolved to perform multiple activities associated with a specific region of the protein. Thus, in order to better understand its functions, we need to thoroughly study its sequence and structure, which, because of its large size and complexity, has not been an easy task. Preliminary analysis of the CagY secondary structure showed that residues 5 to 343 correspond to a 5’ repetitive region domain in the amino terminal (Figure 1a), also known as FRR. The next region (344 to 365) corresponds to a transmembrane region, which may allow anchoring the protein to the IM, although it is unknown if this domain associates with other proteins [13]. Positions 366 to 1458 correspond to a region that includes the middle repeat region (MRR), almost 1000 aa long, sometimes described as a conserved region [13,40]. In fact, it is the most polymorphic region of CagY among different Hp strains [13,40]. Secondary structure predictions of this region show essentially short alpha helix motifs, mainly associated with the repetitive modules (Figure 1a,b). CagY proteins from different Hp strains show changes in the arrangement and number of these modules, affecting the length of CagY [13,40]. It is worth noting that the MRR is the only region of CagY where the amino acid Cys is present, and in an unusual number and distribution (see Figure 4a), e.g., the Hp 26,695 strain has a total of 56 Cys residues, two for each repetitive module. It has been suggested that these residues may be involved in the formation of disulfide bonds, playing an essential role in the folding stability of CagY; they are probably also needed for maintaining the multimer structure, hence affecting the function of CagY as a modulator of the T4SS activity [13,14]. 

The region from residues 1469 to 1927 has been described in different structural studies with cryo-EM and cryo-ET techniques [11,12,23,43]. This is also the only region of CagY that has homology with the VirB10 proteins from other bacterial T4SSs, such as *Escherichia coli* and *Agrobacterium tumefaciens* [12,13,23]. Experimental structures for this region are reported in the RCSB-PDB database (Table 1). However, recent studies have reported an asymmetric complex with 17 chains of CagY/CagX (6X6J) and a symmetric complex with only 14 CagY chains (6ODI), which is known as the symmetry mismatch [11,12]. Previous studies with cryo-ET have detected only a 14-chain structure complex, however [12,23].

Due to the complexity of this protein, its complete structure has not been experimentally elucidated yet. Recent structural works with cryo-ET and cryo-EM have shown an impressive complex assembly for the T4SS complex, with several proteins coded in the Cag PAI, like CagX, CagY, CagT, CagM, and CagD. However, most of the CagY protein structure remains unsolved. Bioinformatic sequence analysis predicts that the C-terminal of this protein corresponds to an intrinsically disordered region (IDR), consistent with aa sequences that promote sequence disorder. This region is on the cytoplasmic side of the Hp IM, and its structure is difficult to predict because of its disordered character and possible interaction with other proteins, like the energetic protein complexes of the T4SS (Appendix A) [25]. Proteins that interact with CagY, according to STRING (Appendix A), include the CagE virB homologue, involved in DNA transfer and required for induction of IL-8 in gastric epithelial cells [44], and CagT or TrwB, an inner membrane nucleoside triphosphate-binding protein [45]. The conjugal transfer protein family TrbF, known to be involved in conjugal transfer, is thought to be part of the pilus required for molecule transfer [46]. Also, CagX is considered a homologue to VirB9 from the VirB/D4 T4S of the plant pathogen *A. tumefaciens*, and is also essential for the translocation of CagA [47].

Starting from the structures obtained in cryo-ET and cryo-EM studies, we built an initial model by homology modeling using the 6ODI and 6X6J structures as templates and considering a multimer structure with only 14 chains of CagY, consistent with cryo-EM studies. A segment of 72 aa between 6ODI and 6X6J, an “empty zone” not solved in the cryo-EM studies, was solved with an ab initio model. This step was relatively simple because most bioinformatic methods for secondary structure prediction, ab initio, and threading showed agreement with an alpha helix conformation.

The symmetry mismatch previously described in cryo-EM studies arises from detecting only 14 CagY chains in the region closer to the OM and 17 units in the PR region, without any evident connection between the 3 CagY chains in excess [12,24]. Authors of the cryo-EM works admitted that it was not possible to explain the cause for such structural discrepancies. Possible explanations include the presence of truncated versions of CagY (and CagX) in the T4SS, conformational isomers of these proteins, or incomplete assemblies of the T4SS [11,12,43]. However, previous cryo-ET studies reported only 14 chains for the same complex [12,23]. Therefore, we decided to predict the CagY structure model using the symmetric 14 chains of CagY. In fact, the obtained model (Figure 2c) showed that the space occupied by the “asymmetric complex zone” was similar, only with a slightly wider space between the chains. 

More challenging for modeling is the zone that includes the MRR, as it corresponds to the most polymorphic and complex zone of CagY. This zone is of particular interest since previous studies associate sequence variations with the modulation of the T4SS activity for CagA translocation, the immunogenicity response, and the phosphorylation of the CagA oncoprotein [5,15,40]. Sequence variations include losses and gains of the repeated modules A and B and single residue substitutions.

Previous studies suggest that the initial T4SS assembly should start by placing CagY and CagX in the bacterial envelope [48] because the multimer structure of CagY and CagX can be remarkably stable and independent of other proteins. CagX is shorter than CagY and seems to interact only with the OM section of CagY. Therefore, we consider as a reasonable assumption to build multimer models without the presence of the CagX protein for the MRR region. There was good agreement on the secondary structure. The AlphaFold2/ColabFold deep learning approach has been recognized as the most confident approach for predicting protein structure, reaching “experimental confidence” in some cases [32,49,50]. However, the method still has limitations for challenging targets, like complex and large proteins and those for which there are not enough templates for modeling, like CagY [51]. When trying to model the whole sequence of CagY, AlphaFold2/ColabFold yielded a structure that was mainly disordered and difficult to align with the known partial structures from cryo-EM studies. Therefore, we first identified the previously determined domain structures and sequence motifs to build hybrid structure models. Recent works have remarked that deep learning approaches may produce highly confident structure models for proteins with repetitive modules [52,53] and disulfide bonds [54]. Thus, we focused on deep learning approaches for modeling the MRR with AlphaFold2/ColabFold. The modeling of this domain as a monomer had a more compact structure than that obtained with the ab initio methods of I-TASSER and Robetta (Appendix A). However, it was not possible to establish a proper orientation with the structure of the remaining part of CagY. The modeling as a multimer was challenging because the large size and complexity of this domain demanded computer power beyond our available capacity. Still, it was possible to estimate a model for the dimer and despite its low confidence values, the dimer model showed a more compact structure oriented parallelly, allowing us to deduce a correct orientation with the remaining part of CagY. Possible causes for the low confidence values are an insufficient refinement of the conformation of individual chains in the dimer, which showed considerable differences (RMSD > 2Å). On the other hand, stereochemical and structural analysis of the monomer and dimer models showed that all the Cys residues were grouped in pairs at the correct distance to participate in disulfide bonds (typically less than 2.5 Å), further supporting the accuracy of the model [55].

We refined the conformation with EMD simulations by building a trimer from two dimer models to optimize the conformation of the central chain. The RMSD behavior of the trimer dynamics was consistent with the hypothesis that the neighboring chains help to maintain the multimer’s conformation. Notably, the chains with only a single neighbor did not stabilize in conformation. Therefore, we consider the conformation of this central chain as the most appropriate for assembling the remaining part of the multimer.

A key observation was the presence of unusually large numbers of Cys in the MRR region (58 in Hp 26,695 strain) separated at regular intervals (Appendix A). It has been suggested that Cys content is correlated with evolution, where prokaryotes have the lowest content and mammals the highest [56]. Small human proteins represent some of the richer Cys proteins, like in metallothionein domain proteins (MTs) or in granulins (GRNs) with over 20% of Cys content [57]. However, we are unaware of other large proteins with such a high number of Cys concentrated in a region of the protein like in the MRR of CagY. Moreover, Cys usually presents as a C-(X)2-C motif, with disordered domains interspersed with Cys. The above suggests that MRR has had a particular evolutionary history, probably different from the rest of CagY protein and different from most cysteine-rich proteins.

Previous studies already reported the presence of Cys in the MRR of CagY, and authors suggested they may form disulfide bonds necessary for the stability of the region [13,14]. The process involved in their formation and the functions of the disulfide bonds in prokaryotes have been less studied than in eukaryotes [54]. Of note, these motifs are commonly found in transmembrane and extracellular proteins [54]. In our study, the structural analysis of the trimer showed evidence of intrachain disulfide bonds but not interchain bonds. Therefore, these bonds do not seem to play an essential role in the quaternary structure but seem to be most relevant for the stability of the complex structure of CagY, particularly in the large MRR region.

It has been previously described that Hp has numerous Cys-rich proteins and that many of them possibly form disulfide bonds. [58]. For many prokaryotic organisms, the enzymes DsbA and DsbB are the most frequently involved in disulfide bond formation [54]. However, these enzymes are absent in Hp, and instead, the enzyme DsbK (HP0231 gene) is present and probably the most involved in this process [59,60]. Cys-rich proteins are reported to be important to respond to oxidative stress. They may then be essential in the response and regulation of inflammation [61]. It is intriguing to suggest that the redox-sensitive potential of Cys residues in the MRR region of CagY may play a role in its well-described regulation of inflammation in animal models [60,62,63]. We propose a scenario where CagY modifies its structure to modulate the translocation of the CagA protein. MRR disulfide bonds may act as a redox switch or mediate the contraction of this protein region to modulate the T4SS function and CagA translocation.

Some studies have suggested that changes in the AP regions and in the MRR are associated with the translocation capacity of the CagA protein, the immunogenic response, and the phosphorylation capacity [34]. The models for the AP obtained in this work by deep learning and homology allow us to propose that the shortening of the AP region may also affect the translocation capacity by altering its interaction with the OM of Hp. Moreover, the decrease in the pore channel’s diameter may obstruct the passage of CagA. It is important to emphasize that the size of the CagA protein is considerably larger than the AP pore, so the translocation process is more complex and may involve major structural changes.

On the other hand, the models calculated by homology tend to fit the dimensions of the template (in this case, the 6ODI structure), which may explain the opposite prediction of the deep learning methods with respect to the pore diameter. Deep learning methods consider a multitude of factors, and it is generally accepted that they model the interactions between the chains more efficiently [32], suggesting that they may offer a more accurate description of the conformational changes of the pore. However, to obtain more precise data, it will be necessary to use other techniques such as molecular dynamics of these complexes, preferentially including the membrane, where our model may be used as a starting point for such simulations.

The similarity of CagY with some contractile proteins as detected by I-TASSER (Appendix A) suggests that MRR has contractile properties, and given the interaction of the disordered region of CagY with the energy complex beneath the IM, its function might be similar to that of flagellar or pilins proteins [64]. We plan to conduct MD studies to better predict conformational changes in the MRR and AP regions and understand their possible associations with the phenotypes produced by CagY variants present in Hp strains of patients with different gastric conditions.

The optimal construction of a CagY model using computational techniques, which includes the 14 chains that have been described in the asymmetric model, is not a simple task due to the large size of this protein. In principle, we have omitted the initial part of it (the first 331 residues) since bioinformatics analyses suggest that this region is mainly disordered, although it is also possible, according to these same studies (Appendix A), that it may be a region that can undergo an order–disorder transition, which undoubtedly encourages its study shortly.

Still, the rest of the protein is too large to calculate an optimized structure, even with decent computational capabilities. For this reason, in this study, we consider dividing this protein into different domains; in summary, we can consider that a part of it, the one that contains the homologous region with VirB10, can be successfully modeled by combining homology and ab initio modeling strategies, taking advantage of the cryo-EM structures that have been recently published. The part corresponding to the MRR constitutes a challenge due to its complexity. However, we consider that the recent advances in deep learning techniques have been satisfactory for the prediction of structures with the characteristics of this domain, including repetitive regions and disulfide bonds. In this way, each domain has been modeled and optimized independently, and the final assembly of the structure appears to be in agreement with the cryo-EM experimental data and other validation tests of the structure, as shown in this article.

However, we understand that one of its limitations lies in the lack of an optimization process for the complete structure. This can be achieved by using a more powerful computing infrastructure or by resorting to other computational strategies, such as coarse-grained molecular dynamics techniques, such as those based on Gaussian network models (GNMs) [65].

The model of the CagY structure proposed In this work may be used to understand how this protein interacts with CagA, DNA, and heptose during their transit through the T4SS, and the importance of the MRR modules in the reported CagY functions. The tridimensional model will also help better understand its interaction with other proteins of the T4SS, such as CagX, CagM, CagT, and CagD, which are part of the core complex of the secretion system (see Figure 7b). Our model suggests that CagY constitutes the tunnel that spans from the inner membrane of Hp to the membrane of the gastric cell in its host. Furthermore, the assembly of the T4SS components illustrates the role of CagY as the “skeleton” of the system around which all proteins assemble.

A better knowledge of the structure and function of CagY will offer the possibility to envision ways to prevent or modulate the pro-inflammatory activity of the T4SS, e.g., by designing molecules that may block the binding of CagY to its human receptor or designing a vaccine with an inactive structure of the protein.

Another limitation of our proposed model is that we did not include the 17/14 CagY asymmetric model previously suggested [12], and considered only the symmetric 14 CagY subunits to build the model. However, with the available data, it is not possible to determine whether the asymmetry corresponds to incomplete versions of CagY or to important conformational topology and arrangements of the CagY multimer.

In conclusion, using a combination of informatics and analytic tools, we provide a structural model for the (almost) complete CagY protein, particularly the complex and highly diverse MRR region, and present evidence of the role of the unusually abundant Cys residues in stabilizing the protein. We also modeled the AP region and provided proof of the role of sequence variants in CagA translocation. Finally, we offer a more detailed model for the assembly of the T4SS focused on the central role of CagY and show the agreement of our model with cryo-ET and cryo-EM studies.

## 4. Materials and Methods

### 4.1. CagY Model Sequence and Structure Models

The amino acid sequence used to model the structure of the CagY protein corresponded to the wild type (WT) strain Hp 26,695 and was obtained from the NCBI database (WP_103414807). The CagY gene structure models 6X6J and 6ODI, obtained by cryo-EM, are available in the Protein Data Bank (PDB) and were used as homology modeling templates.

### 4.2. Determination of Secondary Structure and Transmembrane Regions

PsiPred workbench v4.0 (http://bioinf.cs.ucl.ac.uk/psipred/) was used to predict structural domains and the secondary structure of the CagY protein (accessed on 4 August 2023). The modules for predicting the secondary structure (PsiPred 4.0), sequence disorder (DISOPRED3), and transmembrane regions (MEMSAT-SVM) were employed.

### 4.3. Modeling of the Protein Monomer and Construction of the Multimer

The Modeller v10.3 program [66] was used for homology modeling of the CagY individual chains, covered by 3D models 6X6J and 6ODI, which correspond to parts of the OM and PR regions of the T4SS. These structures were obtained by cryo-EM at atomic resolutions of 3.50 and 3.80 Å, respectively [24]. The 6ODI corresponds to a 14-chain multimer structure composed exclusively of CagY structures (residues X to Y). The 6X6J structure belongs to a 34-chain structure; half correspond to CagY (residues X to Y), and the remaining 17 chains correspond to CagX. Due to the odd number of chains of both CagY and CagX, this multimer is known as the asymmetric complex. The positions of the atoms in the structures were preserved in the models relative to the T4SS structure. Modeling of the missing segments of CagY was performed with SWISS-MODEL [67], I-TASSER [68], and Robetta servers [69].

Also, Modeller’s multi-template scripts for homology modeling were used for the assembly of models derived from homology and segments from ab initio, as described below. The 3D alignment and assembly of the CagY multimers for the OM and PR regions were made with UCSF ChimeraX 1.6.1 [39] using the coordinates of CagY present in the 6ODI and 6X6J structures as templates.

### 4.4. Modeling of the Middle Repeat Region (MRR) of CagY

The AlphaFold2/ColabFold platform was used for the calculation of the conformation of the segment that includes the MRR of the CagY protein. The AlphaFold2/ColabFold-MMseqs2 method [50] was selected using the pdb70 template mode, with the mmseqs2_uniref_env and unpaired-paired options that suggested five models of the structure, which were evaluated using the lDDT confidence coefficient [70]. The MRR segment was also tested for dimer modeling (AB) using the AlphaFold2/ColabFold-Multimer option. To improve the multimer building, a trimer model was generated from the alignment of two identical dimers with ChimeraX (labeled AB and A′B′) through the aligning of the chains B and A′ of the dimers, and posterior remotion of one of the superimposed chains (A′), to produce an ABB′ trimer. The 3D conformation of the trimer structure was optimized using an equilibrium molecular dynamics (EMD) simulation of 10 ns in explicit solvent, and the system was configured using the CHARMM-GUI platform [71]. The trimer model was then immersed in a cubic solvation box with edge dimensions of 102 Å (minimum spacing of 10 Å from the protein surface to the edges of the box), in an aqueous solution neutralizing charges with KCl ions, and Charmm36m was selected as the force field potential [72]. The dynamic optimization was carried out using the NAMD3 program (Phillips, 2020 #61) with a canonical ensemble (isothermal–isochoric ensemble) in combination with an NPT ensemble (isothermal–isobaric ensemble) and a temperature of 303.15 K. The RMSD of the final EMD trajectories was computed with VMD 1.9.4 [73]. A PDB representative for the cluster of structures in the stable trajectory of the trimer central chain (frame 95) was selected with ChimeraX. For validation porpoises, a second EMD simulation of 5.8 ns was run in the same conditions by building a trimer replacing all chains, with the optimized central chain of the previous dynamics. The stereochemical and optimized structural properties of the monomer, dimer, and trimer were analyzed with the PDBsum-EBI server [74].

### 4.5. Final Assembly of CagY multimer

A CagY multimer structure with 14 chains was assembled with models from ab initio, homology modeling, and the central chain (B) of the trimer model obtained by the EMD simulations. CagY coordinates in the 6ODI and 6X6J structures were used as a reference, as described below. The assembly was completed with ChimeraX and a manual edition of PDB files. The final CagY multimer model was assembled comprising amino acids 363 to 1927. Only the intrinsically disordered region (IDR) of the of the protein (residues 1 to 362) was not included.

### 4.6. Modeling of the Antenna Projections (AP) of CagY

We modeled the structure corresponding to the AP sequences described in the work of Tran et al. [34] and built the AP structures described as WT (residues 1763 to 1863), CagY GS20, replacing 20 residues in the AP region with glycine and serine, CagY Xc, replacing the AP region with the AP region of *Xanthomonas citri*, and CagY ΔAP where the complete AP was deleted (Appendix A). These regions were modeled with SWISS-MODEL/DeepView by using the previously built multi-template model and AlphaFold2/ColabFold2-Multimer.

## Figures and Tables

**Figure 1 ijms-24-16781-f001:**
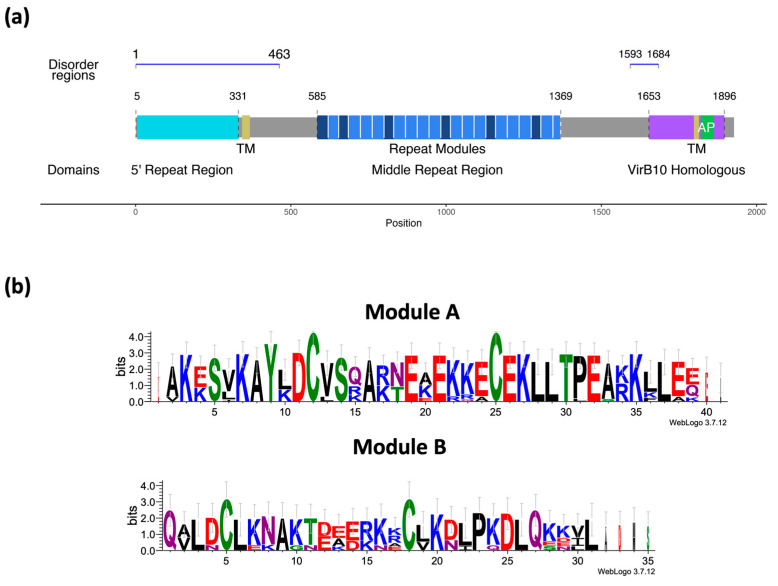
Motifs, secondary structure, and repeated modules of CagY. (**a**) CagY sequence of *H. pylori* 26,695 (WT) is 1927 residues long; structural domains frequently cited are the 5′ repeat region, FRR (cian), the transmembrane regions (TM) (yellow), the middle repeat region (MRR) (blue), the antenna projection (AP) (green), and the VirB10 homologous region (purple). Also, the figure identifies the domains in which the structure was divided for modeling: intrinsically disordered region (IDR), middle repeats region (MRR), and VirB10 homologous region. (**b**) Logos for the sequence of repeated modules A and B from the MRR for the WT strain.

**Figure 2 ijms-24-16781-f002:**
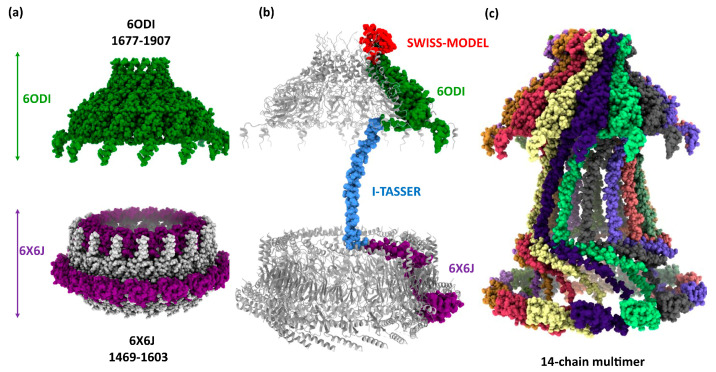
Homology modeling of the VirB10 homologous region. (**a**) PDB structures 6ODI (green) and 6X6J (purple) were used for templates of the OM and periplasmic regions. (**b**) Monomer model (virB10 homology region) and alignment with template structures, 6ODI (green), 6X6J (purple), and the I-TASSER model of residues 1604 to 1677 (blue) and SWISS-MODEL of the AP (red). Complete model calculated with Modeller10v8. (**c**) Multimer 14-chain model of VirB10 homology region. Chains were aligned to the 6ODI structure as a template.

**Figure 3 ijms-24-16781-f003:**
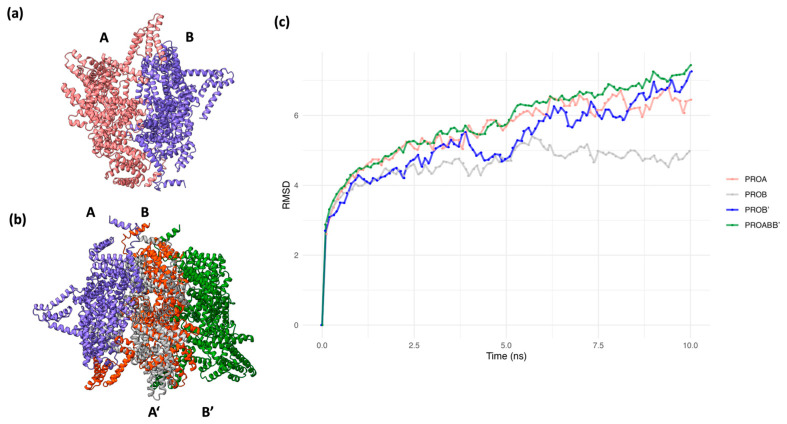
AlphaFold2/ColabFold dimer modeling of MRR, trimer building, and equilibrium molecular dynamics (EMD) for the trimer. (**a**) Best AlphaFold2/Colabfold model dimer structure for the MRR and lDDT values. (**b**) Alignment of two identical dimers, chains labeled as AB (blue and red) and A′B′ (gray and green), respectively, for producing an ABB′ trimer (blue, red, and gray) structure (A′ chain was removed from superposed B/A′ structures). (**c**) RMSD plot of 10 ns molecular dynamics for the trimer. Individual plots for chains A (red), B (blue), and B′ (green) are displayed, as well as the plot for the whole ABB′ trimer. Only chain B stabilized its RMSD values (gray).

**Figure 4 ijms-24-16781-f004:**
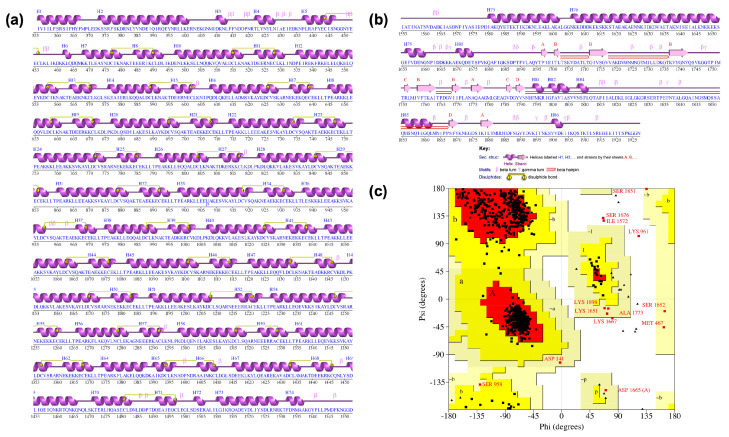
Secondary structure of the CagY Model and Ramachandran plot. (**a**) Secondary structure of the domain, including the MRR (residues 353-1550), where all the 29 disulfide bonds are located. (**b**) Secondary structure of the VirB10 region (residues 1551 to 1927), where no Cys residues are present. (**c**) Ramachandran plot of the CagY assembled multimer, with 95% of the residues in the most favored regions (red), 3.6% in additional allowed regions (yellow), 0.6% in generously allowed regions (light yellow), and 0.3% in disallowed regions (white). Residues with anomalous degree combinations are highlighted in red—all graphics generated by PDBsum.

**Figure 5 ijms-24-16781-f005:**
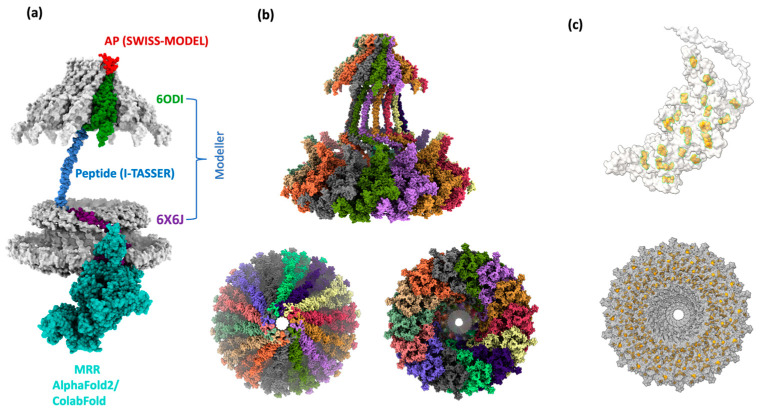
CagY monomer and multimer assembly. (**a**) Summary of the CagY chain monomer assembly with the structure templates and modeling methods indicated. (**b**) Side, bottom, and upper views of the final assembly of the 14-chain multimer. (**c**) Detail of the MRR where all 58 Cys are highlighted in yellow, forming 29 disulfide bonds. The bottom view shows the apparent concentric rings delineated by these bonds.

**Figure 6 ijms-24-16781-f006:**
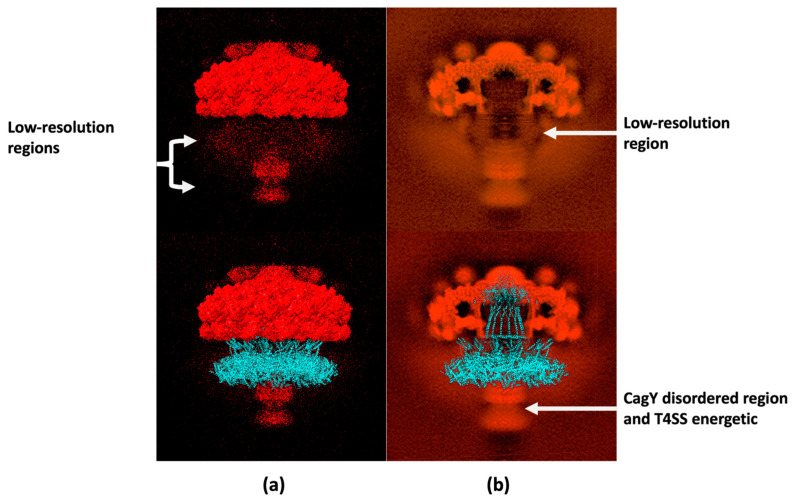
Cryo-EM electrography of the T4SS and location of the CagY modeled multimer. Solid surfaces in the maps correspond to high-resolution regions used to calculate the protein structures. Zones with dots represent low-resolution sections. (**a**) The image at the top shows the surface of the cryo-EM dispersion maps, whereas the bottom diagram displays a superposition of the CagY structure model. There was a good fitting of the CagY model with the “cap” of the cryo-EM surface, which corresponds to the zone of higher resolution, whereas the MRR region of the model occupies the low-resolution portion of the map (in blue). (**b**) A cross-section of the cryo-EM dispersion map where the figure at the top shows the channel of the T4SS, whereas the bottom figure shows the superposition with the CagY model, which fits the cavity and corresponds with the low-resolution zones (3.8 Å resolution cryo-EM maps, EMD accession: EMD-20020).

**Figure 7 ijms-24-16781-f007:**
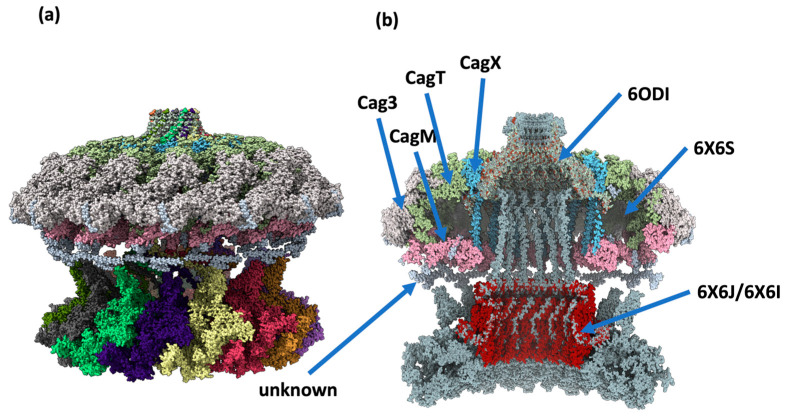
Assembly of the CagY protein with other protein structures of the T4SS from cryo-EM studies. (**a**) The model of CagY fitted with the structure of the T4SS from cryo-EM (6X6S). (**b**) A transversal section of the assembly, showing that the most internal structure corresponds to the CagY multimer (in light grey). Other proteins of the T4SS are displayed in different colors. The structure marked as “unknown” corresponds to a non-identified protein, which is in close proximity to the protuberances of the MRR of the predicted model of CagY.

**Figure 8 ijms-24-16781-f008:**
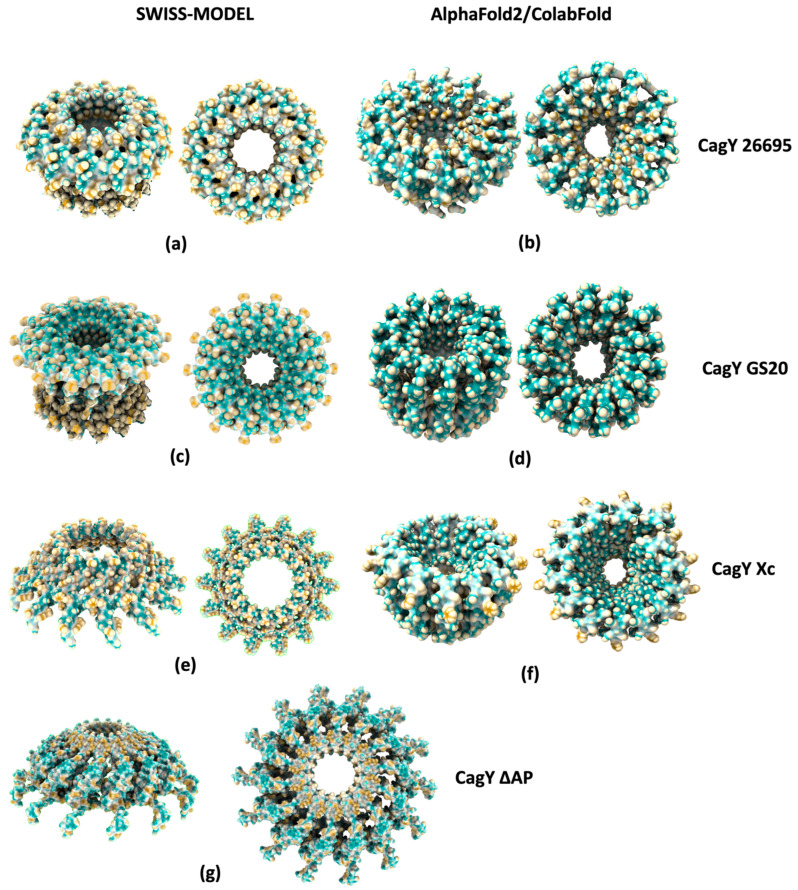
Modeled structures of CagY Antenna projections (AP). (**a**) The wildtype (WT) AP structure modeled by homology shows an open pore with a crest shape. (**b**) The structure of the WT AP of CagY modeled by deep learning shows an open pore with an internal dent. (**c**) GS20 AP structure, with 20 amino acids replaced by Gly and Ser residues, modeled by homology, shows an open pore with a crest shape. (**d**) The AP GS20, modeled by deep learning, displays a slightly deformed open pore. (**e**) The homology model of the AP Xc, where the AP region was replaced with the shorter AP region of *X. citri*, shows a pore with a large diameter. (**f**) The CagY/Xc modeled by deep learning, shows a pore with a small diameter. (**g**) CagY homology model where the AP was completely deleted (DAP), displays a large pore, which lacks all flanking alpha helixes.

**Table 1 ijms-24-16781-t001:** CagY protein regions.

Region	Start	End	Template
FRR	1	331	NA *
TM	343	368	NA
MRR	585	1369	NA
TM	1798	1814	NA
AP	1820	1851	6ODI
VirB10	1653	1896	6ODI and 6X6J

NA *: not available.

## Data Availability

The data found in this work are currently being uploaded in the Model Archive Database.

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
