# Peer review of "A Proposal for a Consolidated Structural Model of the CagY Protein of Helicobacter pylori"

_ijms, 2023, doi:10.3390/ijms242316781_

Round 1

Reviewer 1 Report

Comments and Suggestions for Authors

The study entitled “A proposal for a consolidated structural model of the CagY pro-2 tein of Helicobacter pylori” used various modelling techniques including homology modelling, ab initio, and deep learning to construct a 1595 amino acid chain, revealing a 14-chain CagY multimer structure. While the research holds promise, there is room for improvement, and I suggest the following suggestions to the authors:

Authors need to give more emphasis in Introduction or Abstract on why the complete protein model was generated, and what is the significance of this modelling study.

Please Enlarge the sequence in Figure 1a its not readable properly.

In Table 1, could you clarify the meanings of "inicio" and "fin"? Also, specify the PDB IDs or descriptions for regions with available crystal or cryo-EM structures and clearly outline which regions were modelled in this study, along with the methods employed.

In Figure 2, adding the names of different domains with distinct colour coding would significantly enhance the readability.

I want to see the all the I-TASSER models in Supplementary file. Provide more details how you have selected the best model etc. Furthermore, could you provide information about the VirB10 region in Section 2.2 and present the alignment process in Supplementary Figures for better transparency?

I-Tasser or I-TASSER correct this.

Figre S1 CagY MRR add correct labels for N-C terminals or residue numbers for easy understanding. There is huge difference between the Models predicted by both the servers how authors will justify this.

Bringing Figure 5S into the main manuscript is suggested, and it would also be valuable to validate the model using ProSA web and other pertinent parameters to provide a more in-depth analysis.

If you could kindly provide an explanation as to why the complete model was not subjected to energy minimization before introducing or modelling all 14 subunits, it would offer valuable insight. Additionally, it would be interesting to explore the possibility of enhancing the model's stability through MD simulations.

Add the conclusion or future work section, kindly elucidate how and where researchers can effectively utilize this model and emphasize the significance of the modelled regions for the study. It would be helpful to acknowledge any limitations associated with using segmented crystal structures and clarify whether the model is readily applicable in research or if additional validation and MD simulations are recommended before further use.

Comments on the Quality of English Language

Quality of the English language is fine.

Author Response

Comments and Suggestions for Authors

The study entitled “A proposal for a consolidated structural model of the CagY pro-2 tein of Helicobacter pylori” used various modelling techniques including homology modelling, ab initio, and deep learning to construct a 1595 amino acid chain, revealing a 14-chain CagY multimer structure. While the research holds promise, there is room for improvement, and I suggest the following suggestions to the authors:

1. Authors need to give more emphasis in Introduction or Abstract on why the complete protein model was generated, and what is the significance of this modelling study.

Response: We appreciate the recommendations about the emphasis on the significance of the work. We modify both the abstract and several parts of the introduction, trying to highlight the importance of a more complete structural model of this protein.  

2. Please Enlarge the sequence in Figure 1a its not readable properly.

Response: We modify the Figure 1. As the picture of the Secondary Structure is somewhat complex, we sent it as a supplementary figure (Figure S1). Also, we improved the annotation and captions of both figures.

3. In Table 1, could you clarify the meanings of “inicio” and “fin”? Also, specify the PDB IDs or descriptions for regions with available crystal or cryo-EM structures and clearly outline which regions were modelled in this study, along with the methods employed.

Response:  We corrected the content of the table, replaced the Spanish words, and also included PDB/IDs.

4. In Figure 2, adding the names of different domains with distinct colour coding would significantly enhance the readability.

Response: We added domain names and changed the coloring in Figure 2 in order to improve its readability.

5. I want to see the all the I-TASSER models in Supplementary file. Provide more details how you have selected the best model etc. Furthermore, could you provide information about the VirB10 region in Section 2.2 and present the alignment process in Supplementary Figures for better transparency?

Response: We are including all the I-TASSER models for the peptide that fills the space between the 6ODI and 6X6J regions in the Supplementary Material (Figure S2 ).  For I-TASSER, the best model corresponds to that with the higher C-Score. This C-Score is based on the significance of threading template alignments and the convergence parameters of the structure assembly simulations, and generally in the [-5, 2] range, the higher, the better. Also, we are including in supplementary materials the I-TASSER models, Robetta Models, and the AlphaFold2/ColabFold for the MRR. However, it is important to mention that, due to the discrepancies between the I-TASSER and Robetta models, as well as the fact that in recent years, deep learning methods have had better performance in predicting the structure of proteins with repeated motifs and disulfide bonds (both situations are present in the MRR), we only used the deep learning models for building the CagY final model.

Additionally, we are including as supplementary material the Modeller multi-template alignment modeling method, which was used to build the hybrid template-based 6ODI-I-TASSER-6X6J structure, as well as the script used for modeling (Figure S3).

6. I-Tasser or I-TASSER correct this

Response: We adjusted the name to I-TASSER in the entire manuscript. We also check all the names of the programs in the manuscript for congruence.

7. Figure S1 CagY MRR add correct labels for N-C terminals or residue numbers for easy understanding. There is huge difference between the Models predicted by both the servers how authors will justify this.

Response: Regarding the models for the MRR, neither the I-TASSER nor the Robetta models were acceptable. This is mainly due to the fact that this region is highly repetitive and also includes the region with disulfide bonds. As we highlighted in the discussion, it is now that most of the traditional ab initio methods have poor performance for proteins containing this type of motif. Therefore, we focus on modeling this region with deep learning with AlphaFold2/ColabFold. In recent years, only the deep learning approaches have been efficient for modeling repetitive and disulfide bonds-containing proteins.

Another problem is the vast length of this domain (about 1000 residues). The AlphaFold2/ColabFold models we initially calculated had considerable variation, but the secondary structure was similar, even compared with that from I-Tasser and Robetta Models. Therefore, we suspected that it is necessary to have more time for optimizing the conformation of this structure. For these reasons, we implemented a double strategy to have a more optimal structure for the MMR: (i) Modeling with AlphaFold2/ColabFold as a multimer because the multimer option seems to improve the confidence of the prediction for multimeric proteins. Unfortunately, the multimer option has huge memory and processing requirements, and it was only possible to model a dimer. Therefore, we started from the AlphaFold2/ColabFold dimer model for building a Trimer, which was subsequently optimized by Molecular Dynamics. We rewrite the results and methods in this section to improve clarity.

8. Bringing Figure 5S into the main manuscript is suggested, and it would also be valuable to validate the model using the ProSA web and other pertinent parameters to provide a more in-depth analysis.

Response: We are now including a modified version of Figure 5S in the main manuscript (Which now corresponds to Figure 4). Also, we are including in the Supplementary material validation analysis of the structure calculated with other servers (ProSA, ERRAT, QMEAN4, and QMEANDisCo). Validation values on these servers were acceptable for the model (Supplementary Figures S11 to S15).

9. If you could kindly provide an explanation as to why the complete model was not subjected to energy minimization before introducing or modeling all 14 subunits, it would offer valuable insight. Additionally, it would be interesting to explore the possibility of enhancing the model's stability through MD simulations.

Action: We know the energy minimization of the whole structure is important, but it is a huge structure. Also, due to the big size of even one chain, the minimization may affect the conformation of one individual chain previous to the assembly. The best option for minimizing the energy is applying it to the whole multimer structure,  but this is an even bigger structure.

Moreover, part of the structure is based on templates (the VirB10 homologous region), and the protein we modeled has the same sequence as the molecule used in the structure PDB, cryo-EM models. Therefore we consider that the conformation from these structures should be very proximately maintained. Only the models of the missing domains were included to fill the gaps. For the MRR, the most complex region, a Molecular Dynamics method for optimizing this structure was included. Also, the peptide that fills the gap between the 6odi and 6x6j section was modeled with a multi-template homology modeling with Modeller, using a script that includes an energy minimization strategy (as detailed in the supplementary Figure S3). 

Prior to the assembly, an energy minimization of the whole model seemed inappropriate if it did not consider the interaction with the neighboring chains as it occurs in the multimeric complex.  Conversely, energy minimization of the whole model was not possible due to its big size. However, the model was minimized by domains, and the assembly was careful enough to prevent incorrect contacts and clashes (information included in the manuscript).

We are including information about a second refining of the MRR model by Molecular Dynamics, as it was suggested by the Second Reviewer. Moreover, we plan to conduct other types of Molecular Dynamics studies using coarse-grain methods through Gaussian Elastic Networks, which allow the handling of big molecular complexes, but that will require extensive work by itself. In the meantime, based on the confidence values for the validation, we consider that the model is at least satisfactory based on the available evidence, the methods we were able to conduct in this work, and our available hardware resources.  We add a pair of paragraphs, in the last part of the discussion, about the limitations of the model and possible future works to optimize it (lines 522-545).

10. Add the conclusion or future work section, kindly elucidate how and where researchers can effectively utilize this model and emphasize the significance of the modelled regions for the study. It would be helpful to acknowledge any limitations associated with using segmented crystal structures and clarify whether the model is readily applicable in research or if additional validation and MD simulations are recommended before further use.

Response: We improved the discussion by mentioning some of the limitations of this study and its possible application in research (lines 522-545). 

Reviewer 2 Report

Comments and Suggestions for Authors

The manuscript "A proposal for a consolidated structural model of the CagY protein of Helicobacter pylori" by M. A. López-Luis et al. presents a model of CagY, the largest protein of Type IV Secretion System (T4SS) from Helicobacter pylori (Hp). Although there are structural evidence for large regions of CagY, the authors present a more complete picture of this protein, using homology modeling and other computational tools. In principle, this work is of value and could be interesting for a broad audience of IJMS. However, the manuscript has major drawbacks to be suggested for publication.

To start with, a major issue is the language. There're several grammatical issues that need to be revised. Several words are misused, such as "assignation" in l. 175 instead of assignment and "tridimensional" in l. 191. Legends are also described inefficiently using expressions such as "the down figure" (l. 277). The manuscript should be revised by a professional native English speaker before further revisions.

The computational work has been carried out appropriately for most parts. Below I have some suggestions for clarifications and improvements required:

- The rmsd plot in Fig. 3c does not suggest that the middle chain B is the most representative with lower variability. Rmsd indicates difference from the initial structure, whereas variability can be regarded as flexibility that is measured by atomic fluctuations (rmsf). The central chain was to expected as more stable than the two flanking chains A, C that have less contacts (probably half) than chain B. However, this can be used as a rationale to refine the structure, i.e. run MDs of a trimer, obtain the central optimized chain and build a trimer out of it for a second round of optimization, which should display lower divergence (difference) from the initial structure. This has to be demonstrated.

- Ramachandran plot is a key method of stereochemical analysis of protein residues, however it does not provide quality analysis of the 3D structure of a protein. Therefore, the authors should also present results from other methods such as the z-score analysis from VERIFY3D and ERRAT as well . All these analyses could be conveniently carried out at the SAVES web interface from UCLA.

- Now, the observations drawn about the putative disulfide bonds of MRR domain (l. 210-222) are very interesting (not outstanding as said in l. 222) and should be elaborated with more evidence/experiments from the literature and further analysis of the results obtained. This is done in part within the discussion, therefore the authors should refer the reader to this part.

- The figures are nicely rendered and overall the discussion is good. However, it lacks any potential use of the model presented in drug discovery projects. This is also missing in the introduction and is a key point that should be presented and discussed in there too.

Minor issues:

- Figure 1b is not legible and should be placed with high resolution at full size, preferably within Supplementary Data. Instead the authors could present a schematic representation of the disordered regions or regions of interest for the discussion.

- The numbering shown in Table 1 can be incorporated in Fig. 1a with larger fonts. An alternative Table could be formed by including experimental structures used for each region and the computational methods employed to model each region. 

- "6ODI and 6X6J structures' should be referred to as cryo-EM structures of CagY (PDB IDs: 6ODI and 6X6J), at least in the first sentence (l. 137-138).

- Non-standard abbreviations, such as EMD and IDDT should be explained within the main text, before legends or methods have to be read.

- Figure's 5 legend is not informative and the text next to the arrows could be described in more detail within the legend. Panels and not "up/down figure" should be numbered and explained in detail.

I'm willing to accept a revised version of this manuscript for reviewing, however, I cannot suggest its publication at its present form.

Comments on the Quality of English Language

There're several grammatical issues that should be revised.

Several words are misused, such as "assignation" in l. 175 instead of assignment and "tridimensional" in l. 191. Legend are also described inefficiently using expressions such as "the down figure" (l. 277).

The manuscript should be revised by a professional native English speaker.

Author Response

The manuscript "A proposal for a consolidated structural model of the CagY protein of Helicobacter pylori" by M. A. López-Luis et al. presents a model of CagY, the largest protein of Type IV Secretion System (T4SS) from Helicobacter pylori (Hp). Although there are structural evidence for large regions of CagY, the authors present a more complete picture of this protein, using homology modeling and other computational tools. In principle, this work is of value and could be interesting for a broad audience of IJMS. However, the manuscript has major drawbacks to be suggested for publication.

1. To start with, a major issue is the language. There're several grammatical issues that need to be revised. Several words are misused, such as "assignation" in l. 175 instead of assignment and "tridimensional" in l. 191. Legends are also described inefficiently using expressions such as "the down figure" (l. 277). The manuscript should be revised by a professional native English speaker before further revisions.

Response: We made profound changes to the document to improve the grammar. 

2. The computational work has been carried out appropriately for most parts. Below I have some suggestions for clarifications and improvements required:

The rmsd plot in Fig. 3c does not suggest that the middle chain B is the most representative with lower variability. Rmsd indicates difference from the initial structure, whereas variability can be regarded as flexibility that is measured by atomic fluctuations (rmsf). The central chain was to expected as more stable than the two flanking chains A, C that have less contacts (probably half) than chain B. However, this can be used as a rationale to refine the structure, i.e. run MDs of a trimer, obtain the central optimized chain and build a trimer out of it for a second round of optimization, which should display lower divergence (difference) from the initial structure. This has to be demonstrated.

Response: We appreciate the comments about the computational work. We are including an additional analysis of the RMSD values. Also, we changed the paraphrasing of part of the manuscript. In fact, we have three chains, and we consider that only the central chain in the trimer reaches a state of RMSD equilibrium. Additionally, we are including heatmaps of the RMSD values per residue in the supplementary section (Figure S10a) for the three chains in the simulation.

We consider that the suggestion of running a second MD using the trimmer with the optimized chains is excellent. We ran a second MD of 4.8 ns by building this trimer by substituting the chains with the optimized ones. In this new simulation, we are observing a similar behavior to the previous one, and again, the central chain has less structural fluctuation than the ones at the extremes.  However, the heatmaps for the RMSD values per residue show significantly lower values, especially for the central chain. This seems to indicate that, in fact, this optimized structure is now more stable (Supplementary Figure S10b).  These plots show low flexibility in most parts of the protein (blue zones). Still, there are high variations in the extremes of the protein, which is a relatively common observed phenomenon in Molecular Dynamics (The NAMD3 manual, in fact, recommends ignoring the fluctuations in the RMSD at the extremes of the protein, for global RMSD calculation, as these residues have higher mobility). These results suggest, in general, a slightly more stable conformation of the protein for the central chain and, in general, for the optimized trimer. We are including a brief description of this result in the main manuscript. 

However, we think this system requires a more complex optimization using, at least, the 14-chain multimer. Unfortunately, this is beyond our current computing resources. We additionally include, in the supplementary material, the results of this new molecular dynamics simulation (Supplementary Figure S9). Also, we add to the discusion, some words about possible future work to improve this model.

3. Ramachandran plot is a key method of stereochemical analysis of protein residues, however it does not provide quality analysis of the 3D structure of a protein. Therefore, the authors should also present results from other methods such as the z-score analysis from VERIFY3D and ERRAT as well . All these analyses could be conveniently carried out at the SAVES web interface from UCLA.

Actions:  we are including in the Supplementary material validation analysis of the structure calculated with other servers (ProSA, ERRAT, QMEAN4,  QMEANDISC). Validation values were acceptable for the model.

4. Now, the observations drawn about the putative disulfide bonds of MRR domain (l. 210-222) are very interesting (not outstanding as said in l. 222) and should be elaborated with more evidence/experiments from the literature and further analysis of the results obtained. This is done in part within the discussion, therefore the authors should refer the reader to this part.

Response:  There are few elements in the literature about this observation for CagY as it is, in fact, an unusual protein.  We add a sentence in the results to draw the attention of the reader to the discussion about this.

4. The figures are nicely rendered and overall the discussion is good. However, it lacks any potential use of the model presented in drug discovery projects. This is also missing in the introduction and is a key point that should be presented and discussed in there too.

Response: We are including some words about the potential use of the model in drug discovery. Moreover, we think this model will be useful as a starting point to model other CagY variants from different H. pylori strains, where there has been found some relationship between inflammatory events or immunity with the changes in the structure of the CagY, particularly in the MRR, as well as with the translocation of the CagA oncoprotein through this complex protein. We modified our discussion, trying to highlight these aspects.

Minor issues:

5. Figure 1b is not legible and should be placed with high resolution at full size, preferably within Supplementary Data. Instead the authors could present a schematic representation of the disordered regions or regions of interest for the discussion.

Response: Figure 1 was simplified, and the secondary structure is presented in Supplementary Figure S1.

6. The numbering shown in Table 1 can be incorporated in Fig. 1a with larger fonts. An alternative Table could be formed by including experimental structures used for each region and the computational methods employed to model each region.

Response: We enlarge figure 1a to better appreciate its contents.

7. "6ODI and 6X6J structures' should be referred to as cryo-EM structures of CagY (PDB IDs: 6ODI and 6X6J), at least in the first sentence (l. 137-138).

Response: We include this in the sentence (now in lines 140-142) and in other critical sections of the manuscript where these structures are cited.

8. Non-standard abbreviations, such as EMD and IDDT should be explained within the main text, before legends or methods have to be read.

Response: We check all the standard abbreviations. Some abbreviations were lost/moved because we migrated the "Methods" section to the final part of the document. We checked the paper to fix this issue.

9. Figure's 5 legend is not informative and the text next to the arrows could be described in more detail within the legend. Panels and not "up/down figure" should be numbered and explained in detail.

Response: We improved the description of the content of this figure (now Figure 6).

I'm willing to accept a revised version of this manuscript for reviewing, however, I cannot suggest its publication at its present form.

Comments on the Quality of English Language

There're several grammatical issues that should be revised.

Several words are misused, such as "assignation" in l. 175 instead of assignment and "tridimensional" in l. 191. Legend are also described inefficiently using expressions such as "the down figure" (l. 277).

The manuscript should be revised by a professional native English speaker.

Response: Grammar was significantly improved in the document.

Round 2

Reviewer 2 Report

Comments and Suggestions for Authors

The authors have addressed all issues and comments raised sufficiently. Therefore the revised manuscript is significantly improved, both in terms of language and quality of presentation.